# Insights into the Jasmonate Signaling in Basal Land Plant Revealed by the Multi-Omics Analysis of an Antarctic Moss *Pohlia nutans* Treated with OPDA

**DOI:** 10.3390/ijms232113507

**Published:** 2022-11-04

**Authors:** Shenghao Liu, Tingting Li, Pengying Zhang, Linlin Zhao, Dan Yi, Zhaohui Zhang, Bailin Cong

**Affiliations:** 1Key Laboratory of Marine Eco-Environmental Science and Technology, First Institute of Oceanography, Ministry of Natural Resources, Qingdao 266061, China; 2Laboratory for Marine Ecology and Environmental Science, Pilot National Laboratory for Marine Science and Technology (Qingdao), Qingdao 266061, China; 3National Glycoengineering Research Center, School of Life Sciences, Shandong University, Qingdao 266237, China

**Keywords:** jasmonate, flavonoids, metabolomic profiling, transcriptomic sequencing, bryophytes

## Abstract

12-oxo-phytodienoic acid (OPDA) is a biosynthetic precursor of jasmonic acid and triggers multiple biological processes from plant development to stress responses. However, the OPDA signaling and relevant regulatory networks were largely unknown in basal land plants. Using an integrated multi-omics technique, we investigated the global features in metabolites and transcriptional profiles of an Antarctic moss (*Pohlia nutans*) in response to OPDA treatment. We detected 676 metabolites based on the widely targeted metabolomics approach. A total of 82 significantly changed metabolites were observed, including fatty acids, flavonoids, phenolic acids, amino acids and derivatives, and alkaloids. In addition, the transcriptome sequencing was conducted to uncover the global transcriptional profiles. The representative differentially expressed genes were summarized into functions including Ca^2+^ signaling, abscisic acid signaling, jasmonate signaling, lipid and fatty acid biosynthesis, transcription factors, antioxidant enzymes, and detoxification proteins. The integrated multi-omics analysis revealed that the pathways of jasmonate and ABA signaling, lipid and fatty acid biosynthesis, and flavonoid biosynthesis might dominate the molecular responses to OPDA. Taken together, these observations provide insights into the molecular evolution of jasmonate signaling and the adaptation mechanisms of Antarctic moss to terrestrial habitats.

## 1. Introduction

Oxylipins are a large group of lipid-derived signaling molecules that are produced when polyunsaturated fatty acids undergo oxidation [1,2]. Upon release from membrane lipids by lipases, such as DEFECTIVE IN ANTHER DEHISCENCE1 (DAD1) and DONGLE (DGL), they are oxidized to form hydroperoxides by lipoxygenases (LOX) [2,3]. Jasmonic acid (JA) and (+)-7-*iso*-jasmonoyl-l-isoleucine (JA-Ile), the two core oxylipins, play a significant role in plant defense against pathogens and herbivores. There are also recent studies suggesting that other oxylipins, such as 12-oxo-phytodienoic acid (OPDA), contribute to plant stress responses [2]. The applications of exogenous jasmonates are well demonstrated for their ability to trigger a large spectrum of actions, including seed germination, root growth, trichome initiation, and chlorophyll degradation [4,5]. The presence of jasmonate is also essential for modulating the resistance of plants to insect attack, pathogen infection, wounding, UV radiation, salt stress, and drought stress. Additionally, they are considered to be effective chemical elicitors of secondary metabolites, such as flavonoids and alkaloids [5,6].

One of the most well-understood jasmonate signaling pathways is the transcriptional regulation of JA-responsive genes, which is mediated by the CORONATINE INSENSITIVE1 (COI1) receptor interacting with the jasmonate ZIM-domain (JAZ) protein [7]. By conjugating JA to isoleucine (Ile), JA is activated for signaling. COI1 receptor and JAZ protein interact physically via JA–Ile conjugates [8,9]. Multiple transcription factors are known to interact with JAZ proteins to regulate downstream processes [10]. Transcriptional factors are relieved from JAZ-mediated repression when JAZ proteins are degraded by the 26S proteasomal complex [7]. Bryophytes use OPDA as a ligand to activate a conserved signal receptor that regulates similar jasmonate responses [11,12]. However, OPDA-mediated signaling pathway remains elusive.

*Cis*-(+)-12-oxo-phytodienoic acid, known as *cis*-(+)-OPDA, is the cyclopentenone precursor of JA [3]. Through the octadecanoid pathway, OPDA is synthesized in plastids by oxidizing octadecatrienoic acid (18:3n-3) by 13-lipoxygenase (13-LOX) to produce 13-hydroperoxylinolenic acid. This reaction is then catalyzed by allene oxide synthase (AOS) and allene oxide cyclase (AOC) to produce OPDA [2,3]. OPDA is further transferred from plastids to the peroxisome, leading to JA biosynthesis. These later steps include the reduction of *cis*-(+)-OPDA by OPDA reductase 3 (OPR3), activation to the coenzyme A (CoA) ester, and three cycles of β-oxidation. *Arabidopsis thaliana* harbors at least five genes encoding OPDA reductase but only OPR3 catalyzes the formation of OPDA. *A. thaliana* mutants lacking the OPR3 gene (which belongs to the class OPR II) are resistant to insects and fungal pathogens, demonstrating that *cis*-(+)-OPDA is sufficient to control plant defense responses [13]. Furthermore, OPDA is more than just a biosynthetic precursor of jasmonates [2]. OPDA functions as a signaling molecule having more efficiency at inhibiting seed germination than JA, which is independent of COI1 but synergistic with abscisic acid (ABA) [14].

Bryophytes generally possess most of the key innovations of land plants and are considered the first terrestrial plants [15]. The moss *Physcomitrium patens* and liverwort *Marchantia polymorpha* can synthesize OPDA but lack OPR3 (OPDA reductase) that catalyzes the hormone product of JA–lle in vascular plants [11,16]. The COI1 receptor is functionally conserved in land plants but the ligands it binds are dn-*cis*-OPDA and dn-*iso*-OPDA in *M. polymorpha* and *Calohypnum plumiforme* [16,17]. However, the signal transduction and the endogenous responses involved in OPDA signaling remain elusive in bryophytes [18]. Scientists have sequenced and released several genomes of bryophytes, such as *P. patens*, *M. polymorpha*, *Anthoceros angustus*, *Ceratodon purpureus*, and *Pohlia nutans* [11,19,20,21]. These new genomes and transcriptome sequencing projects provide abundant supports for evolutionary analysis of hormone signaling. In general, bryophyte genomes contain only a minimal but complete set of land-plant signals. For example, *M. polymorpha* contains single orthologs of the COI1 receptor, JAZ protein, MYC transcription factor, and NINJA adaptor to the TPL repressor [21,22,23]. Bryophytes do not synthesize JA–Ile but do accumulate OPDA, so they can provide a useful model to examine how jasmonates affect COI1-independent responses [12]. These highlight the importance of studying OPDA-specific responses and the evolution of jasmonate signaling in bryophytes [2].

Mosses and lichens are the dominant terrestrial vegetation on the Antarctic continent [24,25]. Psychrophilic mosses from Antarctica are emerging models for studying responses and sensitivities to environmental stressors [26]. Our study aims to reveal the global properties of Antarctic moss *P. nutans* as a result of treatment with OPDA using both metabolomics and transcriptomics approaches. A total of 676 metabolites were identified with 82 significantly changed metabolites (SCMs) between OPDA treatment and control groups (45 upregulated and 37 downregulated). Moreover, the representative differentially expressed genes (DEGs) were summarized into different categories, including Ca^2+^ signaling, ABA and JA signaling, lipid and fatty acid biosynthesis, flavonoid biosynthesis, antioxidant enzymes and detoxification proteins, transcription factors, and other stress-related genes. Integrated multi-omics analysis highlights the role of hormone signaling (jasmonate and ABA) and secondary metabolites (lipids and flavonoids) in response to OPDA treatment and the adaptation to terrestrial extreme environments.

## 2. Results

### 2.1. Metabolite Profiling and Significantly Different Metabolites under OPDA Treatment

To reveal potential signaling pathways of *P. nutans* in response to OPDA treatment, we detected the metabolites in a qualitative and quantitative manner using UPLC-MS/MS termed as widely targeted metabolomics technique. First, we analyzed three components of score plots (PC1, PC2, and PC3) in principal component analysis (PCA), calculating 53.8%, 13.98%, and 12.92%, respectively. Two groups were separated (CK and OPDA) (Figure 1A). Thus, PCA score plot demonstrated the high reliability of the experiments. Additionally, samples were clearly divided into two groups, suggesting significant differences in metabolite classes and quantities between two groups. To identify variables contributing to differences between groups, we used the supervised model of orthogonal projections to latent structure-discriminant analysis (OPLS-DA) to compare metabolite contents. We calculated the differences between OPDA and CK (R^2^X = 0.658, R^2^Y = 1, Q^2^ = 0.961) by using the OPLS-DA model (Figure 1B). The Q^2^ value of the OPLS-DA model was larger than 0.9, showing that the evaluation model was stable (Figure 1B). These results indicated that OPDA significantly altered the metabolic profiles of *P. nutans*.

There were 676 metabolites detected in the sample, including 101 phenolic acids, 100 amino acids, 79 organic acids, 73 free fatty acids, 56 saccharides and alcohols, 52 nucleotides, 34 lysophosphatidylcholine (LPC), 31 alkaloids, 28 flavanols, 26 lysophosphatidylethanolamine (LPE), 17 glycerol ester, 15 vitamin, 14 plumerane, 13 flavones, 8 phenolamines, 7 coumarines, 6 flavanones, 2 flavanonols, 2 phosphatidylcholine, 2 sphingolipids, and 10 others (Appendix A). There were 6.84% of flavonoid metabolites, including chalcones, flavones, flavanols, flavonols, flavanones, and flavanoneols. Among them, apigenin, eriodictyol, hesperetin, kaempferol, luteolin, and quercetin were the major intermediate metabolites in flavonoid synthesis pathway, which were all detected in *P. nutans*. Flavonoids, however, mainly appeared as O-linked glycosides, such as apigenin-7-*O*-rutinoside, hesperetin-5-*O*-glucoside, luteolin-7-*O*-glucuronide, kaempferol-3-*O*-rutinoside, and quercetin-3-*O*-glucoside (Appendix A).

We found 82 significantly changed metabolites (SCMs), with 45 upregulated and 37 downregulated, between the treatment and control with the criteria of |log_2_Fold Change| ≥ 1 and variable importance in project (VIP) ≥ 1 (Figure 1C, Appendix A). According to the values of log_2_(Fold change), the top 20 SCMs were identified and are shown in a graphical plot (Figure 1D). Among them, 3-Hydroxy-3-methylpentane-1,5-dioic acid belonged to amino acids and derivatives and was the most markedly accumulated compound with log_2_(Fold Change) of 15.97. In addition, 3-Methyl-2-oxobutanoic acid, a kind of organic acid, was the second markedly accumulated metabolite with log_2_(Fold change) of 13.34. Lipids and flavonoids are the main differential metabolites (Figure 1D). Lipids accounted for 42.68 % of the total SCMs, whereas flavonoids accounted for 29.58 % (Figure 2A and Appendix A). In the class of lipids, palmitic acid, 2-linoleoylglycerol-1,3-di-*O*-glucoside, 1-linoleoylglycerol-2,3-di-*O*-glucoside, and 1-linoleoyl-sn-glycerol-diglucoside were the most accumulated metabolites under OPDA treatment, while methyl linolenate, lysoPE 20:3, 9-oxo-12z-octadecenoic acid, 15(*R*)-hydroxylinoleic acid, and lysoPC 16:2 were the significantly reduced metabolites (Figure 2A).

To uncover the functions of SCMs, we conducted the KEGG enrichment analysis (Figure 2B). A hypergeometric test’s *p*-value was used to determine the significance of enriched pathways. In general, the more significant the enrichment is, the closer the *p*-value is to 0. KEGG enrichment analysis found that DCMs were mostly abundant in pathways including linoleic acid metabolism, flavonoid biosynthesis, flavone and flavonol biosynthesis, alpha-linolenic acid metabolism, arachidonic acid metabolism, and caffeine metabolism (Figure 2B). In order to measure metabolic correlation between significantly different metabolites, Pearson correlation analysis was used to measure synergistic or mutually exclusive relationships. A chord diagram was made for paired differential metabolites according to the screening threshold of |r| ≥ 0.8 and *p* < 0.05. There are higher metabolic proximities between flavonoids and phenolic acids but lower metabolic proximities between lipids and phenolic acids (Figure 2C).

### 2.2. Transcriptome Sequencing and Differentially Expressed Genes under OPDA Treatment

We used transcriptome sequencing to determine the global gene expression profiles of *P. nutans* treated with OPDA. Raw reads were filtered to remove low-quality reads to improve data quality and reliability. The number of raw reads, clean reads and clean bases, the ratio of Q_20_/Q_30_, and the percentage of GC in each dataset were summarized (Appendix A). Six samples yielded a total of 38.36 Gb clean reads after quality trimming. The Q_30_ percentage for all libraries was greater than 93.66%, and the average GC content was 52.12%. We conducted the sequence alignment for mapping clean reads to the *P. nutans* genome. The Pearson’s correlation coefficient (R^2^) between samples within each group was approximately equal to 1, indicating a higher degree of reliability in subsequent differential gene analyses (Figure 3A). To discover the overall gene expression differences, we drew a boxplot to reflect the FPKM distribution. The median FPKM values of OPDA treatment groups were lower than those of control groups, showing that OPDA application repressed the global gene expression (Figure 3B). Nevertheless, maximum values of OPDA treatment groups were higher than those of control groups, demonstrating that several genes were specifically upregulated under OPDA treatment. Our RNA-sequencing results demonstrate that OPDA is an effective regulator for *P. nutans* gametophytes, as well as a high-quality bioinformatics analysis.

We further accessed the global profiles of differentially expressed genes (DEGs), which were screened based on the threshold of adjusted *p*-value < 0.05 and |log_2_(foldchange)| > 1. A hierarchical clustering analysis of DEGs revealed that samples were clearly separated into two groups, indicating that OPDA treatment and control samples differ significantly in their classes and quantities (Figure 3C). Consequently, a total of 4816 DEGs were identified between OPDA treatment and control (Appendix A). Of them, 3131 DEGs were upregulated and 1686 DEGs were downregulated (Figure 3D). To reveal the biological functions of DEGs, we performed GO enrichment analysis. In the biological process, DEGs were summarized to coping with abiotic stresses, oxylipin or jasmonate biosynthetic and metabolic processes, ion transport, metabolic process, and homeostasis. In the molecular function, DEGs were mainly involved in ion transporter activity, channel activity, transmembrane transporter activity, hydrolyzing *O*-glycosyl compounds, and secondary active transporter activity. In the cellular component, DEGs were predominantly distributed into plant-type vacuole, intrinsic component of plasma membrane, and plant-type vacuole membrane (Figure 4A). Furthermore, the number of enriched genes, rich factor, and q-value were used to analyze KEGG pathway enrichment. There were several pathways with high enrichment, such as metabolic pathways, starch and sucrose metabolism, biosynthesis of secondary metabolites, glycerophospholipid metabolism, alpha-linolenic acid metabolism, and amino acid metabolism (Figure 4B). The representative DEGs were further classified into multiple classes, such as calcium signaling pathway, abscisic acid signaling pathway, jasmonate signaling pathway, antioxidant enzymes and detoxification proteins, lipid and fatty acid metabolic pathway, flavonoid synthesis and metabolic pathways, carbohydrate metabolism and glycosyltransferase, and transcription factors (Appendix A). Our findings suggest that differentially transcriptional regulation of genes reflects fine-tuning responses to OPDA.

### 2.3. The Crosstalk between OPDA and ABA Signaling Pathways

The function of jasmonate is always dependent on complex crosstalks with several other hormonal signaling pathways. Previous publications have uncovered some clues about the crosstalk between jasmonate and ABA. To comprehensively reveal the signal interaction, we used heatmaps to show the expression levels of the core component genes in OPDA and ABA signaling pathways obtained from the transcriptome sequencing (Figure 5A). OPDA treatment triggered the upregulation of genes involved in jasmonate biosynthesis and signaling transduction, including lipoxygenase (*LOX*), allene oxide synthase (*AOS*), allene oxide cyclase (*AOC*), 12-oxophytodienoic acid reductase (*OPR*), and *JAZ* protein. Meanwhile, OPDA increased the expression levels of genes in ABA signaling, such as pyrabatin resistance/pyrabatin resistance-like/regulatory components of ABA receptor (*PYR*/*PYL*/*RCAR*), type 2C protein phosphatase (*PP2C*), sucrose nonfermenting 1-related protein kinase 2 (*SnRK2*), and abscisic acid insensitive 5 (*ABI5*) (Figure 5A and Appendix A). In addition, we confirmed the upregulated expression of these genes under OPDA treatment by qPCR analysis, including *AOS* (Poh0285230.1), *AOC* (Poh0370410.1), *OPR* (Poh0051570.1), *PYL* (Poh0172290.1), *PP2C* (Poh0036960.1), *SnRK2* (Poh0360030.1 and Poh0016430.1), *ABI5* (Poh0324020.1), and *JAZ* (Poh0024230.1 and Poh0281560.1) (Figure 5B). Based on previous observations [27,28,29,30], we proposed a schematic diagram of the core component interaction between jasmonate and ABA signaling pathways (Figure 5C). Therefore, the regulation mechanism of OPDA stimulates the gene expression of jasmonate and ABA signaling was evolutionarily conserved among land plants, which contribute to coping with a variety of abiotic stresses.

### 2.4. Integrated Transcriptomic and Metabolomic Analysis Highlights Glycerophospholipid Metabolism Pathway and Differentially Expressed Genes

Phospholipid biosynthesis is a core metabolic pathway that strongly influences plant growth and development, as well as stress responses. In eukaryotic membranes, phosphatidylethanolamine (PE) and phosphatidylcholine (PC) are the two most abundant phospholipids. By referring to the KEGG database, we outlined the basic phospholipid biosynthetic pathway containing PC, PE, lysophosphatidylcholine (lysoPC), and lysophosphatidylethanolamine (lysoPE) (Figure 6A). Plants lack a mutant model for the PC biosynthetic pathway, making its role in plant growth and development unclear. In addition to its role as a metabolite of phospholipids, lysophospholipids have also been reported more and more as intracellular signaling molecules. In the present study, we found that most of LysoPC (i.e., LysoPC 18:4, LysoPC 19:1, LysoPC 20:0, LysoPC 20:2(2n isomer), LysoPC 20:4, and LysoPC 20:5) were markedly accumulated under OPDA treatment (Figure 2A). We used heatmaps to show the expression levels of the genes involved in phospholipid biosynthesis that were obtained from the transcriptome sequencing (Figure 6A). Most of the genes involved in phospholipid biosynthesis were upregulated under OPDA treatment (Appendix A). Further, qPCR analysis confirmed the upregulated profiles of several selected genes, including *CHO* (Poh0084220.1), *PISD* (Poh0018780.1), *PLA2G* (Poh0120240.1), *GDE1* (Poh0050830.1 and Poh0219590.1), *EPT1* (Poh0323340.1), *PEMT* (Poh0307150.1), and *CPT1* (Poh0348200.1) (Figure 6B). Free fatty acids accounted for 28.05 % of the total significantly different metabolites (Figure 2A). Except Palmitic acid, 1-Linoleoyl-sn-glycerol-diglucoside, and 13-Methylmyristic acid, other fatty acids were all markedly downregulated under OPDA treatment (Figure 2). We also identified lots of DEGs (i.e., 3-ketoacyl-CoA synthase, 3-ketoacyl-CoA thiolase, and long-chain acyl-CoA synthetase) involved in fatty acid biosynthesis catalyzing carbon chain elongation in *P. nutans* (Appendix A).

### 2.5. Integrated Transcriptomic and Metabolomic Analysis Highlights Flavonoid Biosynthesis Pathway and Differentially Expressed Gene

We found that *P. nutans* was capable of synthesizing flavones (apigenin and luteolin), flavanones (eriodictyol and hesperetin), flavonols (kaempferol and quercetin), dihydroflavonols (dihydroquercetin and dihydromyricetin), and flavonoids (epicatechin, epigallocatechin, and catechin) (Appendix A). By referring to previous publications [31], we present a model for flavonoid biosynthesis in *P. nutans* (Figure 7A). Among DCMs, the content of most flavonoids was increased under OPDA treatment (Figure 2A). In plants, flavonoids usually exist in the form of *O*-glycosylated derivatives. The *O*-glycosylated derivatives accounted for 52.17% of total significantly changed flavonoids, including kaempferol-3-*O*-sophoroside, quercetin-3-*O*-glucoside, and hesperetin-7-*O*-rutinoside. We used heatmaps to illustrate the expression levels of flavonoid synthase genes in transcriptome sequencing. We retrieved the markedly upregulated or downregulated genes involved in flavonoid biosynthesis, including four *CHS*, two *CHI*, seven *F3′H*, fourteen *2-OGD* (*F3H*/*FLS*/*FNS*), and eighteen DFR from the transcriptome sequencing data (Figure 7A). In addition, qPCR confirmed that OPDA treatment upregulated the expression of genes, including *CHS* (Poh0234360.1 and Poh0239650.1), *CHI* (Poh0039230.1 and Poh0178020.1), *F3′H* (Poh0205200.1 and Poh0004720.1), *2-OGD* (Poh0056150.1 and Poh0225840.1), and *DFR* (Poh0007930.1 and Poh0251360.1) (Figure 7B). Based on our findings, we hypothesized that OPDA might promote adaptation to extreme environments by upregulating genes and accumulating flavonoids.

## 3. Discussion

### 3.1. OPDA Is a Signaling Molecule in the Basal Land Plants

Jasmonates are plant-specific oxylipins that comprise a major defense hormone class in plants. Unlike vascular plants, bryophytes (i.e., *P. patens* and *M. polymorpha*) produce OPDA, but not JA, in response to wounding and pathogenic infection [11,12,18]. OPDA is more than just a jasmonate precursor and has been demonstrated to exert both COI1-independent and COI1-dependent biological activities [2,32]. In addition, exogenous application of OPDA, dn-OPDA, or dn-*iso*-OPDA induces the transcription of JAZ transcripts in bryophytes [17,22,23]. OPDA, dn-OPDA, and their isomers function as forceful signal molecules to induce the signaling downstream of COI1-JAZ in bryophytes [17,22]. Particularly, lots of genes and pathways are preferentially induced by OPDA or Ile-conjugate of OPDA (OPDA–Ile) in a COI1-independent manner [33]. To reveal the biological function of OPDA in basal land plants, we employed an integrated multi-omics approach to investigate the global alterations in metabolites and global transcriptional profiles of an Antarctic moss *P. nutans*. A total of 676 metabolites were detected using the widely targeted metabolomics technique. In total, 82 SCMs were identified under OPDA treatment, which were involved in several biosynthesis pathways, including fatty acids, flavonoids, alkaloids, amino acids and derivatives, and phenolic acids (Figure 1 and Figure 2). In addition, the transcriptome sequencing showed that the identified DEGs were summarized into the function, including Ca^2+^ signaling, abscisic acid signaling, jasmonate signaling, lipid and fatty acid biosynthesis, transcription factors, antioxidant enzymes and detoxification proteins, and other proteins related to biotic and abiotic stresses (Figure 3 and Figure 4). Previously, transcriptomic and proteomic analyses have been conducted to understand the molecular basis of OPDA signaling in *P. patens*. The proteomic analysis of *P. patens* treated with OPDA revealed that the most affected proteins were related to photosynthesis, protein synthesis, and metabolism [34]. In *P. patens*, OPDA also activated allene oxide cyclase (PpAOC1), demonstrating positive feedback regulation [34]. Abiotic stress mainly increased proteins involved in photosynthetic activity, amino acid metabolism, redox equilibrium, energy synthesis, and chaperonin synthesis 24 h after wounding [35]. In the liverwort *M. polymorpha*, OPDA stimulates a COI1-independent signaling that induces thermotolerance genes and protects plants against heat stress [12]. The OPDA signaling pathway plays a critical role in terrestrial adaptation and in coping with current climate change [12]. These highly indicated that the first half of the octadecanoid pathway is conserved and that OPDA can function as a signaling molecule in the basal land plants.

### 3.2. The Crosstalks between OPDA and ABA Signaling Pathways Highlight the Complex Interactions among Core Components

The function of ABA signaling in plants is well understood and it plays a critical role in mediating growth and stress responses. The core ABA signal transduction involves PYR/PYL/RCAR (the ABA receptors), type 2C protein phosphatase (PP2C), sucrose nonfermenting 1-related protein kinase 2 (SnRK2s), and abscisic acid-responsive elements (ABFs) [36,37]. There have been some clues regarding crosstalk between JA and ABA based on previous observations. Exogenous application of JA increased the gene expression in ABA biosynthesis in rice, and these two hormones might work together to increase the expression of transcription factors *WRKY*, *MYC2*, and *bZIPs* during rice defense [38]. In the present study, we also found that OPDA treatment triggered the upregulation of core component genes in OPDA and ABA signaling pathways (Figure 5A,B). By regulating *COI*, *JAZ*, and *MYC2* transcripts, ABA receptors participate in jasmonate signaling for the chlorophyll degradation and anthocyanin biosynthesis [29]. ABA receptor NtPYL4 is involved in the metabolic reprograming of tobacco for alkaloid production induced by MeJA [30]. These findings elucidate that ABA receptor is a connection between the core components of ABA and JA signaling pathways. The JAZ protein and MYC2 are also key components in the crosstalk between JA signaling and other plant hormone signaling that regulates plant growth and stress responses [39]. Through interactions with abscisic acid insensitive 5 (ABI5), Arabidopsis JAZ protein negatively modulates ABA responses and inhibits ABI5 transcriptional activation [27]. This result provides particular insights into the crosstalk between jasmonate and ABA signaling pathways in the mediation of seed germination [14]. Abscisic acid insensitive 4 (ABI4) integrates jasmonate and ABA signals to effectively regulate cold tolerance in apple cells via the JAZ–ABI4–ICE1–CBF signaling cascade [28]. The function of OPR3 and JAR1 genes are possibly missing in basal land plants, the result being that *P. patens* and *M. polymorpha* can only synthesize the JA precursor OPDA. Except OPR3, other OPDA reductases are likely involved in plant stress resistance. The wheat OPR gene (*TaOPR1*) can enhance the tolerance of plants to salt stress through upregulation of *MYC2* transcripts, and then activate the ABA signaling pathway [40]. We also found that the OPR gene family is largely expended in the Antarctic moss *Pohlia nutans* [41]. In the present study, three OPR genes were upregulated by the treatment of exogenous OPDA. However, it is still uncertain whether there is an OPR that possesses catalytic activity and for which compounds were subsequently synthesized after OPDA biosynthesis in bryophytes. We proposed a schematic diagram of the core component interaction between jasmonate and ABA signaling pathways in bryophytes (Figure 5C). Taken together, these findings implied that the key components in JA and ABA pathways might have comprehensively interacting and regulatory mechanisms.

### 3.3. OPDA Differentially Regulates the Biosynthesis of Fatty Acids and Phospholipids

Lipids include triglycerides, which are made up of three fatty acids (FAs) and one glycerol molecule. Fatty acids are key factors to regulate fluidity and stability of cell plasma membrane. Very-long-chain fatty acids (VLCFAs) have hydrocarbon chains longer than 18 carbon atoms, mainly in the form of glycerol phospholipid, sphingomyelin, triacylglycerol, and wax [42]. In the plastid, saturated and monounsaturated C16 and C18 fatty acids are elongated and exported to the cytosol, where they are activated as acyl-CoAs. Long-chain acyl-CoAs are the immediate precursors of VLCFA and are further elongated by FAE complexes within the endoplasmic reticulum [43]. In the present study, we found that most of C18 fatty acids and VLCFAs (i.e., 12-oxo-5,8,10,14-eicosatetraenoic acid, 13*S*-hydroperoxy-9*Z*,11*E*-octadecadienoic acid, 15-oxo-5*Z*,8*Z*,11*Z*,13*E*-eicosatetraenoic acid, and 2*R*-hydroxy-9*Z*,12*Z*,15*Z*-octadecatrienoic acid) were markedly downregulated under OPDA treatment (Figure 2). The carbon chain length, unsaturation, and head structure polarity of VLCFAs make them extremely diverse in structure and function and play a key role in transmembrane transport, cell division and differentiation, energy storage, resistance to stress, and other physiological processes [42,44]. Our result showed that lots of markedly upregulated or downregulated family genes (i.e., 3-ketoacyl-CoA synthase, 3-ketoacyl-CoA thiolase, and long-chain acyl-CoA synthetase) involved in phospholipids biosynthesis in *P. nutans* (Figure 6 and Appendix A). The regulatory mechanism of OPDA on the biosynthesis of VLCFAs remains unclear. Bryophytes will be one of the best choices to reveal the OPDA action on the FAE complexes and VLCFAs biosynthesis.

Phospholipids are the main components of membrane lipid bilayer, including glycerol phospholipids and sphingomyelin [45]. They are synthesized de novo by the Kennedy pathway. Membrane phospholipids have a glycerol backbone, polar headgroup, and two hydrophobic chains, which provide a template for a wide range of chemical species [46]. Lysophospholipids were produced from phospholipids by releasing a long hydrophobic carbon chain under the action of phospholipase, which includes lysophosphatidylcholine (LPC), lysophosphatidylethanolamine (LPE), lysophosphatidylserine (LPS), lysophosphatidylglycerol (LPG), lysophosphatidic acid (LPA), and lysophosphatidylinositol (LPI). In the present study, we found that most LPCs (i.e., LysoPC 18:4, LysoPC 19:1, LysoPC 20:0, LysoPC 20:2(2n isomer), LysoPC 20:4, and LysoPC 20:5) were markedly accumulated under OPDA treatment (Figure 2). PC is the most abundant phospholipid in eukaryotic cell membranes and LPC serves as a lipid signaling precursor or as a ligand for regulatory proteins [47].

### 3.4. OPDA Induces the Biosynthesis of Flavonoids Adapting to Terrestrial Environment

In land plants, flavonoids serve as signal molecules and antioxidants under stress conditions and have prominent roles regulating plant growth and development [48,49]. Flavonoids, such as flavonols and anthocyanins, emerged sequentially as land plants evolved from aquatic to terrestrial environments, facilitating their adaptation to harsh conditions [19]. However, the molecular process of flavonoid biosynthesis in bryophytes and basal land plants is largely unknown [50]. In the present study, flavonoids, including flavanols, flavanones, flavones, and flavonols, accounted for 7.25% of the total compounds in *P. nutans* (Appendix A). Under stress conditions, bryophytes can synthesize early colorless UV-B ultraviolet-absorbing substances rather than the down-stream products, such as anthocyanins [19,51]. Flavone and flavonol biosynthesis, however, belong to the early metabolites of flavonoid pathway, and they are catalysed by enzymes of different branching pathways into anthocyanins [52]. We draw an outline of the biosynthesis pathways of flavonoids for *P. nutans* according to the compounds detected by UPLC-MS/MS analysis (Figure 7). The most abundant compounds were flavonols comprised of O-linked glycosylated quercetin and kaempferol (Appendix A). Flavonols, such as kaempferol, quercetin, and myricetin, are most studied for their antioxidant properties and potential role in UV protection through UV-screening function [48].

Jasmonates have been uncovered for their most prominent function in plants, that is, to regulate the biosynthesis of secondary metabolites, including anthocyanins, artemisinin, glucosinolates, isoquinolines, nicotine, terpenoids, and indole alkaloids [5,6]. Moreover, bryophytes can only synthesize the precursor OPDA and cannot further synthesize jasmonic acid. However, there are few publications about plant responses to OPDA using multi-omics technologies, particularly in the basal land plants. In the present study, flavonoids were the main compounds that accounted for 29.58 % of the total significantly changed metabolites in OPDA treatment vs. CK in *P. nutans* (Figure 2, Appendix A). The MBW transcriptional control complex, composed of MYB, bHLH, and WD40, plays a central role in the modulation of flavonoid biosynthesis [52]. Jasmonates can regulate the biosynthesis of anthocyanins through the interaction between JAZ protein and WD-Repeat/bHLH/MYB transcription complex [53]. Consequently, the dynamic changes in flavonoid constituents caused by OPDA treatment might facilitate moss fight against ROS-induced damage and diverse biotic and abiotic stresses.

Particularly, we found that most upregulated flavonoids were the O-linked glycosylated flavonoids under OPDA treatment (Figure 4). Flavonoids can be modified by glycosylation, resulting in changes in solubility, stability, and chemical functions, while deglycosylation plays a crucial role in regulating flavonoids’ homeostasis [54]. There is likely to be a reserve pool of flavonoid glycosides, which may be easily mobilized when growth conditions are unfavorable [54,55]. Our results showed that OPDA treatment also resulted in significant accumulation of gallate-derived compounds (e.g., gallocatechin 3-*O*-gallate, epicatechin gallate, and catechin gallate) (Figure 4). These gallate-derived compounds usually have strong antioxidant activities. Generally, epicatechin gallate and epigallocatechin-3-gallate are obtained from green tea, which act as powerful antioxidants, preventing oxidative stress caused by hydrogen peroxide and radicals [56,57,58,59].

## 4. Materials and Methods

### 4.1. Plant Materials and Growth Conditions

*Pohlia nutans* was sampled from the vicinity of the Great Wall Station in the Fildes Peninsula of Antarctica (62°13.260′ S, 58°57.291′ W). The plants were grown on a medium of Pindstrup substrate and normal soil (weight ratio 1:1) in flowerpots at 16 °C and 50 µmol photons·m^−2^·s^−1^ light with 16-h light/8-h dark photoperiod conditions. The plastic film that covers mosses retains moisture. For plant hormone treatment, the plants were sprayed with 10 μmol/L OPDA (Item No. 88520, Cayman Chemical, Ann Arbor, MC, USA) for 6 h. Control plants were those without OPDA treatments. We immediately froze the moss gametophytes with liquid nitrogen after collecting them. Three biological replicates were collected and subjected for transcriptome sequencing and LC-MS/MS analyses.

### 4.2. Widely Targeted Metabolome Analysis

The moss gametophytes from OPDA treatment group (i.e., OPDA) and control group (i.e., CK) were used for metabolite analysis. According to previously described methods, metabolite extraction, qualitative and quantitative analyses were conducted [49,60,61]. Briefly, the samples were freeze-dried and ground to powder using a Mixer Mill MM400 (Retsch, Haan, Sachsen-Anhalt, Germany). A total of 0.2 g of powder from each sample was then extracted with 1.2 mL of 70% aqueous methanol at 4 °C overnight. During overnight incubation, the mixtures were vortexed six times to ensure a successful extraction. For detecting the metabolites, the extracts were centrifuged for 10 min at 10,000× *g* and the supernatant was filtered and used for UPLC-MS/MS analysis. We employed the ultra-performance liquid chromatography (UPLC) (Shim-pack UFLC SHIMADZU CBM30A, Kyoto, Japan) and tandem mass spectrometry (MS/MS) (SCIEX QTRAP 6500, Applied Biosystems, Framingham, MA, USA) to carry out a widely targeted metabolome analysis [49]. In addition to the public database of metabolites, the self-construct MWDB V2.0 database (5000+ compounds, Metware Biotechnology Co., Ltd. Wuhan, Hubei, China) was used to determine the primary and secondary spectral properties of metabolites. With triple quadrupole mass spectrometry, metabolites were quantified through multiple reaction monitoring. To monitor reproducibility of detection, a quality control (i.e., mix) was prepared by mixing extract samples.

A multivariate data analysis for the raw data signals was conducted using the supporting software (SCIEX, Framingham, MA, USA). The OPDA-induced accumulation of metabolites was analyzed using principal component analysis (PCA), orthogonal projections to latent structure-discriminant analysis (OPLS-DA), and hierarchical clustering analysis (HCA) using the R package (www.r-project.org/, accessed on 12 July 2022) [60,62]. We conducted a permutation test (200 permutations) for OPLS-DA to avoid overfitting. The multiple changes in metabolites (fold change) were log-transformed to normalize the data. OPLS-DA model was used to calculate variable importance in projection (VIP) values for each metabolite to determine their relative importance. We used the threshold of absolute log_2_(fold change) ≥ 1 and VIP ≥ 1 to determine the significantly changed metabolites (SCMs) between OPDA treatment and control groups. SCMs were annotated and mapped using KEGG compound and pathway databases (www.kegg.jp, accessed on 12 July 2022).

### 4.3. High-Throughput Sequencing and Transcriptome Profiling

Total RNA was isolated from each sample using the TRIzol reagent (Invitrogen, Carlsbad, CA, USA). We determined RNA purity and concentration using Agilent bioanalyzer 2100 (Agilent Technologies, Santa Clara, CA, USA) and monitored RNA degradation using 1% agarose gel electrophoresis. Following the manufacturer’s instructions, sequencing libraries were constructed using NEBNext^®^ UltraTM RNA Library Prep Kit for Illumina^®^ (New England Biolabs, Ipswich, MA, USA). Briefly, we collected and purified mRNA from total RNA using magnetic beads. mRNA was fragmented in the Fragmentation Buffer and cDNA was synthesized by reverse transcription reaction. The synthesized cDNA with ligated adaptor in 250~300 bp length was further purified using AMPure XP system (Beckman Coulter, Brea, CA, USA). Finally, cDNA libraries were constructed from cDNA by PCR amplification using Phusion High-Fidelity DNA polymerase. The quality of each library was detected on the Agilent Bioanalyzer 2100 (Agilent Technologies, Santa Clara, CA, USA).

Using the TruSeq PE Cluster Kit and the cBot Cluster Generation System (Illumina, San Diego, CA, USA), the index-coded samples were clustered according to the manufacturer’s manual. Using the Illumina Hiseq platform, the libraries were sequenced (each with raw data of 6G), generating paired-end reads. The raw reads of FASTQ format were first filtered using fastp program v0.19.3 to remove reads with adapters, reads containing poly-*N* (N content > 10%), and low-quality reads (Q_phred_ < 20). The high-quality clean reads were then retrieved and used for following analyses. Firstly, the clean reads were aligned to the reference genome of *P. nutans* [19], using HISAT program v2.1.0 to generate a bam format file with index information [63]. Then, we assigned sequence reads to genes and acquired the read summarization using featureCounts program v1.6.2 [64]. Finally, the gene expression levels were calculated by the expected number of fragments per kilobase of transcript sequence per million base pairs sequenced (FPKM). The significance of gene differential expression between OPDA treatment and control was estimated using the DESeq2 v1.22.1, and the adjusted *p*-value was assessed using the Benjamini and Hochberg method. The differential expression genes (DEGs) were discriminated with the threshold of adjusted *p*-value < 0.05 and |log_2_(Foldchange)| > 1. BLAST alignment was performed against the Swiss-Prot and NR databases to identify gene functions, and GOseqR was used to analyze gene ontology (GO) enrichment. KEGG pathway analysis was performed using KOBAS software and enrichment analysis was performed based on the hypergeometric distribution test (Mao et al., 2005).

### 4.4. Quantitative Real-Time PCR Technique for the Gene Expression Analysis

In order to validate the expression levels of DGEs in transcriptome sequencing, we performed a quantitative real-time RT-PCR analysis (qPCR). The mosses were sprayed with 10 μmol/L OPDA for 6 h, 12 h, and 24 h. Control plants were those sprayed with sterile water. The gametophytes were cut and frozen with liquid nitrogen. The total RNA was isolated from moss gametophytes using EasyPure^®^ Plant RNA Kit (Transgen, Beijing, China). Then, the first-strand cDNA was synthesized using 0.5 μg of total RNA with TransScript^®^ All-in-One First-Strand cDNA Synthesis SuperMix for qPCR with One-Step gDNA Removal Kits (Transgen, Beijing, China). To normalize the template, several candidate genes were evaluated for their expression stability under OPDA treatment. *GAPDH* was selected as the best reference gene with the most stable expression. The gene-specific primers for qPCR analysis are listed in Appendix A. Finally, we performed the qPCR analysis on a LightCycler96 qPCR instrument (Roche, Basel, Basel-Stadt, Switzerland) with PerfectStart^®^ Green qPCR SuperMix Kits (Transgen, Beijing, China). In this process, the cycling regime was 94 °C for 30 s, followed by amplification for 40 cycles (94 °C for 10 s, 57 °C for 15 s, and 72 °C 10 s). We calculated the relative expression levels of each gene and plotted the histograms using the comparative Ct (2^−ΔΔCt^) method [65]. Three biological replicates were conducted for the experiments.

### 4.5. Statistical Analysis

Three biological replicates were conducted in all experiments and data were presented as the mean ± SD. By using one-way ANOVA test, we calculated the statistical significance of the results between groups in quantitative real-time PCR analysis (** p* < 0.05, *** p* < 0.01).

## 5. Conclusions

We conducted an integrated multi-omics analysis to reveal global properties of the Antarctic moss *P. nutans* under OPDA treatment. We confirmed that the first half of the octadecanoid pathway is conserved and OPDA can function as a signaling molecule in bryophytes. Exogenous application of OPDA can not only activate the jasmonate signaling, but also the ABA signaling, supporting the complex interactions among core components and crosstalk between these two signaling pathways. We found that several lysophosphatidylcholine, O-linked glycosylated flavonols, and gallate-derived compounds were significantly accumulated under OPDA treatment. We proposed that these metabolites induced by OPDA might function as signal molecules and antioxidants protecting against reactive oxygen species (ROS). These results will enlarge our understanding about the evolutions of hormone signaling and the adaptations of these basal land plants to the polar terrestrial environments.

## Figures and Tables

**Figure 1 ijms-23-13507-f001:**
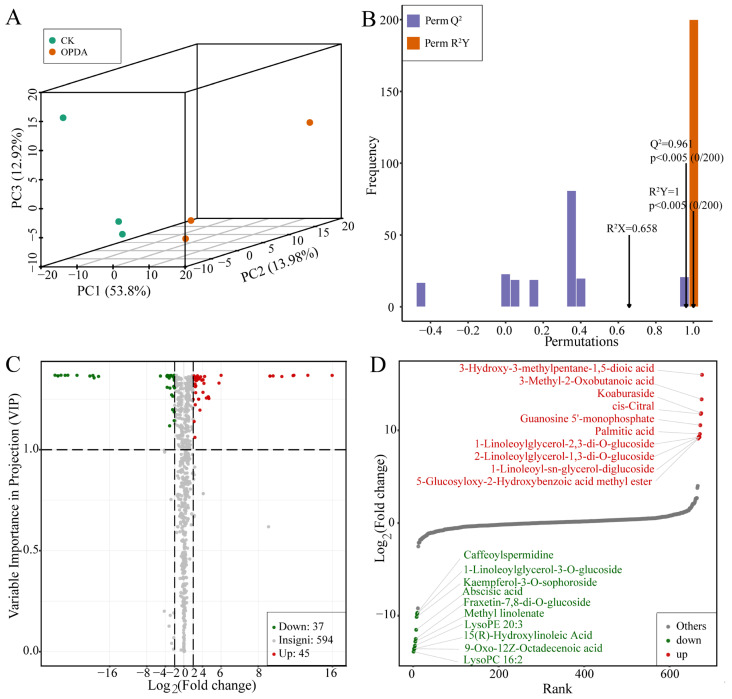
Identification of differently accumulated metabolites in *Pohlia nutans* under OPDA treatment. (**A**) Principal clustering analysis for OPDA treatment and control groups. CK, control group; OPDA, OPDA treatment group. (**B**) OPLS-DA was used to supervise and calculate variables responsible for group differences. R^2^X and R^2^Y represent the interpretation rate of X and Y matrix, respectively, while Q^2^Y indicates the prediction ability of the model. The model is more stable and reliable if the value is closer to 1. Additionally, a model with Q^2^Y > 0.5 is considered effective, and a model with Q^2^Y > 0.9 is considered excellent. (**C**) Volcanic plot showing metabolite contents and statistical significance. Each point represents a metabolite. A horizontal ordinate represents the fold change in metabolites between two groups, while a VIP value indicates a significant difference in statistical analysis. (**D**) The top 20 differently changed metabolites between two groups were selected with the values of fold change.

**Figure 2 ijms-23-13507-f002:**
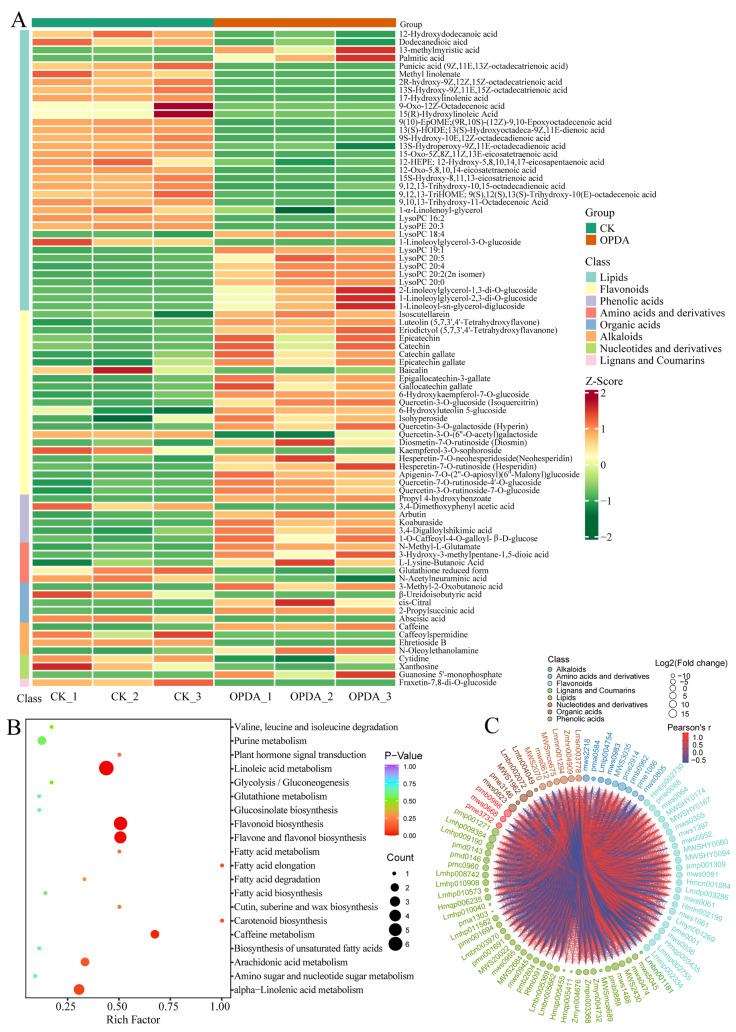
Cluster analysis of differentially changed metabolites in *Pohlia nutans* under OPDA treatment. (**A**) Heatmap showing the differentially changed metabolites (DCMs) sorted by metabolite classes under OPDA treatment. (**B**) KEGG enrichment analysis of DCMs between two groups. Rich factor represents the ratio of the DCMs amount to the total amount of annotated metabolites in the pathway. (**C**) Chord diagram showing the correlation among DCMs identified by Pearson correlation analysis. The outer layer is the metabolite identifier. The point size in the middle layer represents the log_2_(Fold Change) value. The inner line reflects the Pearson correlation coefficient (r) between metabolites. Positive correlation is represented by the red line, while negative correlation is represented by the blue line. The differential metabolite pairs with |r| ≥ 0.8 and *p* < 0.05 were charted.

**Figure 3 ijms-23-13507-f003:**
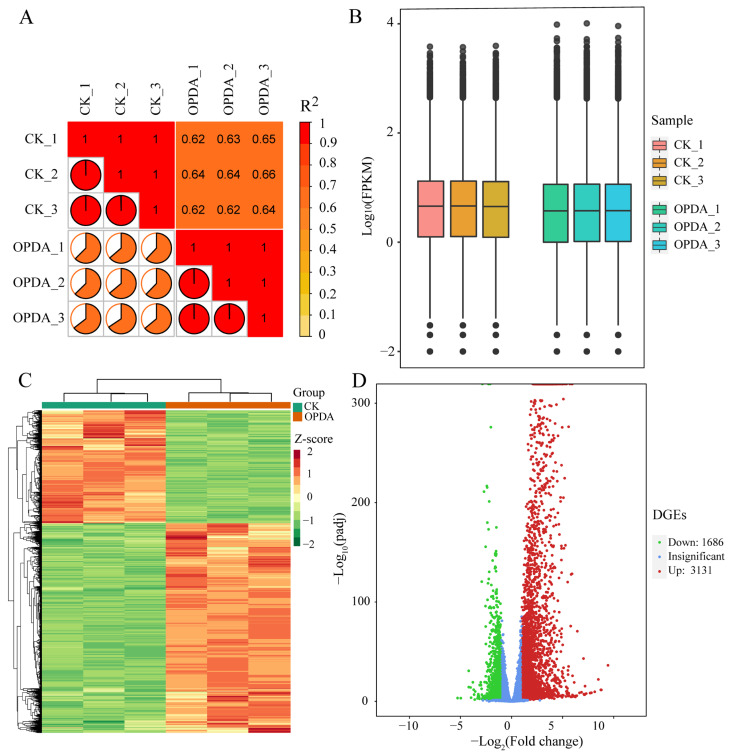
Transcriptome sequencing analysis of *Pohlia nutans* under OPDA treatment. (**A**) RNA-sequencing data from different samples were compared using Pearson correlation coefficient (r^2^) to evaluate parallel levels in each group. The closer r^2^ is to 1, the stronger the correlation between the two duplicate samples. The number in charts and the orange or red ratio in pie charts both represent the value of the correlation coefficient (r^2^) between the two samples. (**B**) The box diagram compared the gene expression levels in different samples. There were five statistics for each region (top-down: maximum, upper quartile, median, lower quartile, and minimum), and the width of each box indicates the number of genes. (**C**) Hierarchical clustering analysis (HCA) for differentially expressed genes (DEGs). In clustering heatmap, horizontal ordinate shows the sample name and vertical ordinate indicates the DEGs. The dendritic plot to the left of the heatmap represents DEGs clustering after HCA analysis. (**D**) The volcano plot showing the DEGs between different groups. X-axis indicates fold change (threshold, |log_2_(Fold Change)| > 1), while Y-axis means the statistically significant level (threshold, log_10_(padj) > 1.3).

**Figure 4 ijms-23-13507-f004:**
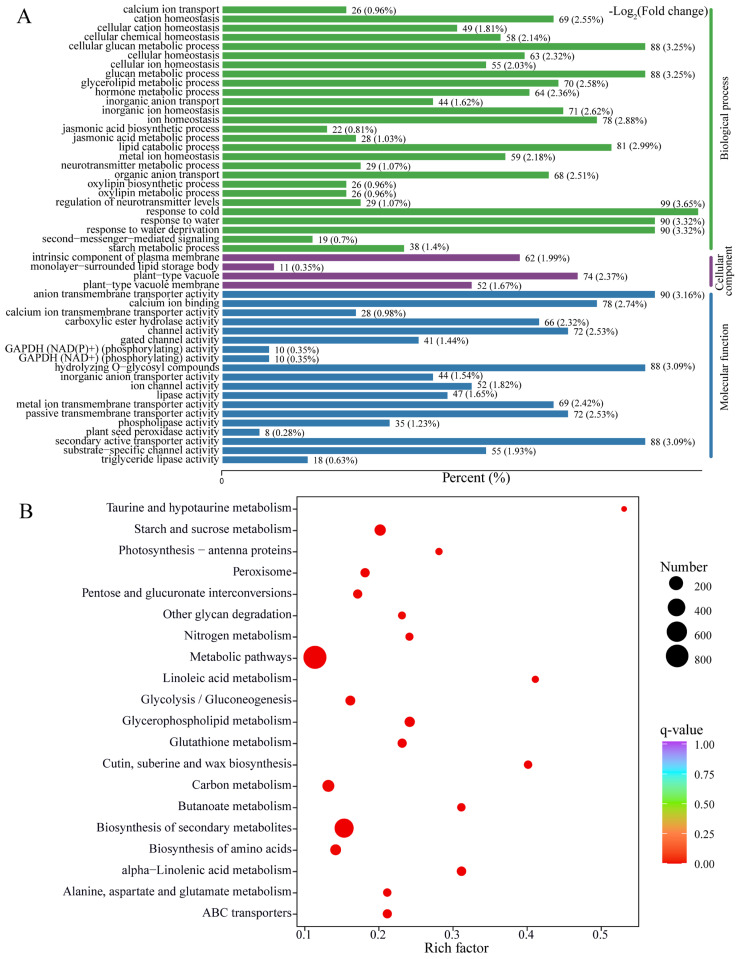
Functional annotation of differentially expressed genes. (**A**) GO enrichment analysis of differentially expressed genes (DEGs). (**B**) KEGG pathway enrichment of DEGs. Gene ratio is calculated as the ratio of DEGs to all annotated genes.

**Figure 5 ijms-23-13507-f005:**
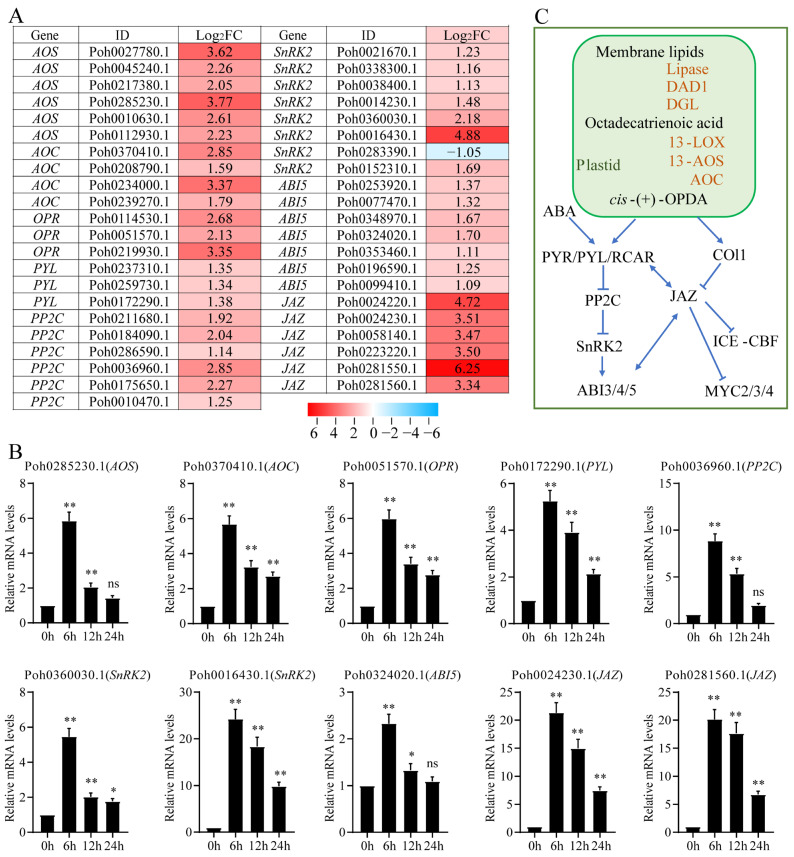
Genes involved in jasmonate and ABA signaling pathways were upregulated under OPDA treatment. (**A**) Heatmap showing that the core components in jasmonate and ABA signaling were markedly upregulated under OPDA treatment in transcriptome sequencing. (**B**) qPCR analysis confirmed that several genes involved in jasmonate and ABA signaling were significantly upregulated under OPDA treatment. *AOS*, allene oxide synthase; *AOC*, allene oxide cyclase; *OPR*, 12-oxophytodienoic acid reductase; *ABI5*, abscisic acid insensitive 5; *JAZ*, jasmonate ZIM-domain protein; *PYR*/*PYL*/*RCAR*, pyrabatin resistance/pyrabatin resistance-like/regulatory components of ABA receptor; *PP2C*, type 2C protein phosphatase; *SnRK2*, sucrose nonfermenting 1-related protein kinase 2. Significant difference (* *p* < 0.05, ** *p* < 0.01, ns, no significant). (**C**) A proposed model summarizing the crosstalk between jasmonate and ABA signaling pathways.

**Figure 6 ijms-23-13507-f006:**
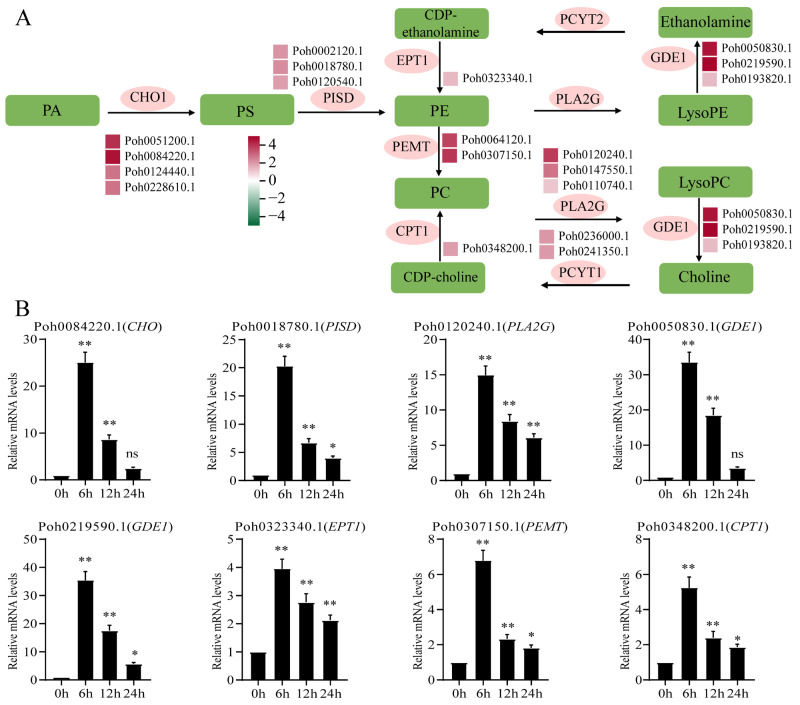
Glycerophospholipid metabolism pathway and differentially expressed genes. (**A**) The basic phospholipid biosynthetic pathway with heatmap showing gene expression levels. The left block of each gene indicated the log_2_(fold change) under OPDA treatment obtained from transcriptome sequencing analysis. (**B**) qPCR analysis confirmed that several genes involved in phospholipid biosynthesis were markedly upregulated under OPDA treatment. *CHO1*, CDP-diacylglycerol-serine *O*-phosphatidyltransferase; *PISD*, phosphatidylserine decarboxylase; *PLA2G*, secretory phospholipase A2; *PEMT*, phosphatidylethanolamine/phosphatidyl-N-methylethanolamine *N*-methyltransferase; *GDE1*, glycerophosphodiester phosphodiesterase; *PCYT1*, choline-phosphate cytidylyltransferase; *CPT1*, diacylglycerol cholinephosphotransferase. Significant difference (* *p* < 0.05, ** *p* < 0.01, ns, no significant).

**Figure 7 ijms-23-13507-f007:**
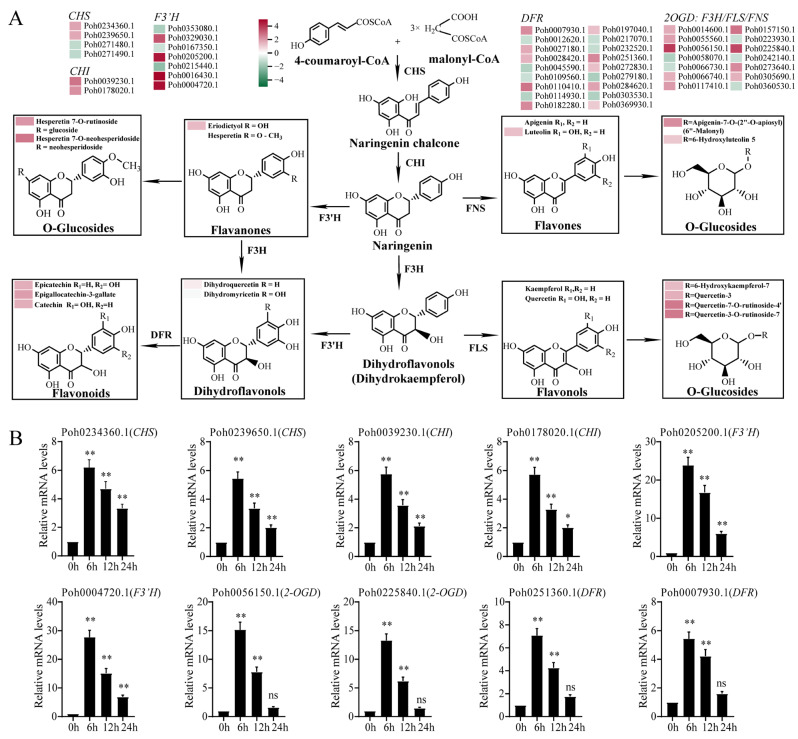
Integrated multi-omics analysis reveals the role of flavonoid pathway under OPDA treatment. (**A**) Proposed biosynthetic pathway of flavonoid synthesis in *Pohlia nutans* with heatmap showing gene expression levels. The color box of each gene indicates the log_2_(fold change) obtained from transcriptome sequencing analysis. *CHS*, chalcone synthase; *CHI*, chalcone isomerase; *F3H*, flavanone 3-hydroxylase; *FLS*, flavonol synthase; *FNS*, flavone synthase; *2-OGD*, 2-oxoglutarate/Fe(II)-dependent dioxygenase; *F3′5′H*, flavonoid 3′-hydroxylase; *DFR*, dihydroflavonol 4-reductase. (**B**) qPCR analysis confirmed that several genes involved in flavonoid biosynthesis were markedly upregulated under OPDA treatment. Significant difference (* *p* < 0.05, ** *p* < 0.01, ns, no significant).

## Data Availability

The reference genome sequence data were available in the National Genomics Data Center (NGDC, https://ngdc.cncb.ac.cn, accessed on 16 September 2022) under the BioProject number PRJCA008231. The transcriptome sequencing data were deposited in Genome Sequence Archive (GSA, https://ngdc.cncb.ac.cn/gsa/, accessed on 16 September 2022) under the BioProject number PRJCA012262.

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
