# Peer review of "Insights into the Jasmonate Signaling in Basal Land Plant Revealed by the Multi-Omics Analysis of an Antarctic Moss Pohlia nutans Treated with OPDA"

_ijms, 2022, doi:10.3390/ijms232113507_

Round 1

Reviewer 1 Report

Title: Insights into the jasmonate signaling in basal land plant revealed by the multi-omics analysis of an Antarctic moss Pohlia nutans treated with OPDA

Summary: the proposed manuscript deals with of the impact of OPDA treatment on Pohlia nutans, an Antarctic moss using both metabolomics and transcriptomics approaches.

Broad comments:

Materials and methods section is well described, as well as the results part with several supporting figures.

However, I deplore the low number of biological replicates, resulting in unreliable statistical results.

The authors indicate that they have carried out targeted and quantitative metabolomics analyses but it is not clearly mentioned that they have the standards for the 676 metabolites detected. Only “databases” is mentioned. On the other hand, in tables S1 and S2 it is a question of “level”. Are these metabolite annotation levels? In this case, it must be understood that a certain number of metabolites are putative, in which case the statements and conclusions must be moderated. And would the terms targeted and quantitative still be appropriate?

Specific comments:

Results:

-        Line 187: I don't understand figure 3A with a mix of numbers and pie charts.

-        Line 250: I don't understand the time scale in Figure 5B. In the materials and methods section, only one sampling time is mentioned (same for the figures 6B and 7B).

-        Line 252: gene OPR doesn’t appear on diagram Figure 5C.

-        Figure 7A: wouldn't there be an extra oxygen on the structure of the flavonol glycosides?

Material & methods:

-        Lines 473-475: are the growing conditions representative of the environment in Antarctica where the moss comes from?

-        Line 562: Can you justify the use of a parametric test of Student. Have you checked the normality of your data? With only 3 replicates per modality, wouldn't a non-parametric test be more appropriate?

-        Line 578: a legend is necessary for table S1. In particular, could the authors describe the different annotation levels. Same for the other tables.

Reviewer 2 Report

In this manuscript, the authors made a multi-omic study to elucidate the role of OPDA in a basal plant; and their precursor roles for jasmonate and interlinked communication with ABA signaling. The work is well-made, and the obtained data made their conclusions possible. 

I found just details of writing that the author could correct and made a careful revision along the manuscript.

  1. The word detected is duplicated on lines 18 and 19, complicating the phrase.
  2. In line 41, when describing the effects of jasmonate, it seems out of context "male fertility" and "reproduction." Since the other terms refer to plants but, in these cases, are ambiguous.
  3. In line 484, there might be an incorrect citation to Wang et al.; it is unclear if it refers to the reference before or after.

Author Response

1.  The word detected is duplicated on lines 18 and 19, complicating the phrase.

Reply: Thanks. We have corrected it.

2.  In line 41, when describing the effects of jasmonate, it seems out of context "male fertility" and "reproduction." Since the other terms refer to plants but, in these cases, are ambiguous.

Reply: Thanks. We have deleted these ambiguous words.

3.  In line 484, there might be an incorrect citation to Wang et al.; it is unclear if it refers to the reference before or after.

Reply: This is an identification error from Endnote software. I have checked that the reference refers to the content. I have carefully proofread the format of all references.